# Transcriptome Analysis of Co-Cultures of THP-1 Human Macrophages with Inactivated Germinated *Trichophyton rubrum* Conidia

**DOI:** 10.3390/jof9050563

**Published:** 2023-05-12

**Authors:** Bruna Aline Cantelli, Gabriela Gonzalez Segura, Tamires Aparecida Bitencourt, Mariana Heinzen de Abreu, Monise Fazolin Petrucelli, Kamila Peronni, Pablo Rodrigo Sanches, Rene Oliveira Beleboni, Wilson Araújo da Silva Junior, Nilce Maria Martinez-Rossi, Mozart Marins, Ana Lúcia Fachin

**Affiliations:** 1Biotechnology Unit, University of Ribeirão Preto-UNAERP, Ribeirao Preto 14096-900, Brazilmariheinzen@gmail.com (M.H.d.A.); mofazolin@gmail.com (M.F.P.); rbeleboni@unaerp.br (R.O.B.);; 2Department of Biochemistry and Immunology, Ribeirão Preto Medical School, University of São Paulo, Ribeirao Preto 14049-900, Brazil; 3Department of Genetics, Ribeirão Preto Medical School, University of São Paulo, Ribeirao Preto 14096-900, Brazilwilsonjr@usp.br (W.A.d.S.J.);; 4National Institute of Science and Technology in Stem Cell and Cell Therapy, Center for Cell-Based Therapy, Ribeirao Preto 14049-900, Brazil; 5Medicine School, University of Ribeirão Preto-UNAERP, Ribeirao Preto 14096-900, Brazil

**Keywords:** dermatophytes, deep infection, RNA sequencing, LPS, IL-32

## Abstract

Although most mycoses are superficial, the dermatophyte *Trichophyton rubrum* can cause systemic infections in patients with a weakened immune system, resulting in serious and deep lesions. The aim of this study was to analyze the transcriptome of a human monocyte/macrophage cell line (THP-1) co-cultured with inactivated germinated *T. rubrum* conidia (IGC) in order to characterize deep infection. Analysis of macrophage viability by lactate dehydrogenase quantification showed the activation of the immune system after 24 h of contact with live germinated *T. rubrum* conidia (LGC). After standardization of the co-culture conditions, the release of the interleukins TNF-α, IL-8, and IL-12 was quantified. The greater release of IL-12 was observed during co-culturing of THP-1 with IGC, while there was no change in the other cytokines. Next-generation sequencing of the response to *T. rubrum* IGC identified the modulation of 83 genes; of these, 65 were induced and 18 were repressed. The categorization of the modulated genes showed their involvement in signal transduction, cell communication, and immune response pathways. In total, 16 genes were selected for validation and Pearson’s correlation coefficient was 0.98, indicating a high correlation between RNA-seq and qPCR. Modulation of the expression of all genes was similar for LGC and IGC co-culture; however, the fold-change values were higher for LGC. Due to the high expression of the IL-32 gene in RNA-seq, we quantified this interleukin and observed an increased release in co-culture with *T. rubrum*. In conclusion, the macrophages-*T. rubrum* co-culture model revealed the ability of these cells to modulate the immune response, as demonstrated by the release of proinflammatory cytokines and the RNA-seq gene expression profile. The results obtained permit to identify possible molecular targets that are modulated in macrophages and that could be explored in antifungal therapies involving the activation of the immune system.

## 1. Introduction

Dermatophytoses are fungal infections caused by pathogenic fungi that occur in animals and particularly in humans. These diseases affect about 900 million people worldwide [1] and are called dermatophytosis [2,3]. According to data from the World Health Organization (2019), dermatophytes affect about 25% of the world population [4] and 30% to 70% of adults are asymptomatic carriers of these diseases. Epidemiological studies have shown an incidence of dermatophytes of 18.2% in the cases analyzed and 80% of these fungal isolates were characterized as *Trichophyton rubrum* [5]. This species was the predominant etiological agent in 268 positive cases of dermatophytosis [6]. Most dermatophytoses are superficial but may become deep infections when associated with a poor general health status of the patient [7]. Infection begins with fungal adherence, followed by the formation of hyphae that can spread through the tissue, releasing fungal enzymes and other pathogenic factors that mobilize keratinocytes, destroy the epidermal barrier, and increase fungal proliferation. These events trigger activation of the host’s immune response [8], including the modulation of genes involved in the response to fungal infection, in the release of fungal virulence factors, and in the induction of inflammatory cascades [9].

Patients with a weak immune system often develop deep dermatophytosis and, although less frequent, this infection can be fatal. Once the infection is established, the pathogen is able to reach deep layers, such as the dermis and hypodermis, and spread through the lymph nodes and bloodstream [10].

These infections are also reported in patients with a compromised immune system, such as HIV carriers and patients with diabetes mellitus [9,11,12], demonstrating the importance of the immune system in controlling these pathologies. Studies in the literature have shown immunocompromised patients with recurrent dermal abscesses caused by fungi [13,14].

In view of the problems associated with deep infections and the need for discovering new types of treatment in order to control these infections, it is necessary to understand the immunological mechanisms involved in this infection [15]. It is believed that the dermatophyte *T. rubrum* is capable of circumventing the innate immune response of the host cell so that the pathogen is not recognized by the defense system, facilitating the infectious process [16]. Dermatophytes are able to alter their transcriptome in response to the host’s natural defenses, particularly invading keratinized tissues and causing superficial infections that can become chronic and deep [17].

During interaction, the host recognizes highly conserved fungal molecules (e.g., β-glucans), which are components of the cell wall of fungi, such as dermatophytes. These molecules are responsible for stimulating the production of proinflammatory cytokines and chemokines. In superficial dermatophytosis, fungi stimulate keratinocytes that secrete cytokines to attract inflammatory cells. In deep infection, β-glucan is recognized by Toll-like receptor and dectin 1 [18], pattern recognition receptors (PRRs) are responsible for detecting pathogens [19].

Our research group has already described the transcriptional profile of human THP-1 microRNAs involved in fungal infection caused by *T. rubrum* [19]. However, analysis of the transcriptional mRNA profile of this interaction by RNA-seq, which is a recent tool used to investigate the details of pathogen-immune cell interactions that would allow accurate and comprehensive gene expression screening [20], has not been reported in the literature. Therefore, the aim of the present study to perform molecular characterization of deep infection caused by *T. rubrum* using as a tool co-culture with a human macrophage cell line (THP-1).

## 2. Materials and Methods

### 2.1. Human THP-1-Derived Macrophages, Media and Growth Conditions

The human monocytic cell line THP-1 (ATCC TIB202), derived from an acute monocytic leukemia cell line, was purchased from Cell Lines Service GmbH (Eppelheim, Germany). The cells were cultured in an RPMI medium (Sigma Aldrich, St. Louis, MO, USA) supplemented with 10% fetal bovine serum at 37 °C in a humidified atmosphere containing 5% CO_2_. Antibiotics (100 U/mL penicillin and 100 μg/mL streptomycin) were added to the medium to prevent bacterial contamination. THP-1 monocytes were adjusted to 1 × 10^6^ cells/mL in a hemocytometer and differentiated into macrophages using 12.5 ng/mL phorbol 12-myristate 13-acetate (PMA) dissolved in dimethyl sulfoxide (DMSO) in RPMI medium for 24 h at 37 °C in a humidified atmosphere containing 5% CO_2_. After PMA induction, THP-1 cells changed morphology and adhered to the culture dish [21,22].

### 2.2. Trichophyton rubrum Strain, Media, Growth Conditions and Inactivation

*Trichophyton rubrum* CBS 118892 (CBS-KNAW Fungal Biodiversity Center), sequenced by the Broad Institute (Cambridge, MA, USA), was cultured on Sabouraud dextrose agar (Oxoid, Hampshire, England) for 15 days at 28 °C. Conidia were prepared as described previously by [23]. A solution of *T. rubrum* (1 × 10^7^ conidia/mL) was cultured in 5 mL liquid Sabouraud medium for 7 h under gentle shaking, as described in [24,25]. After this period, the fungal material was centrifuged for 10 min at 4000 rpm. Part of these live conidia was stored and the other part was washed with sterile saline and incubated for 60 min at 56 °C for inactivation [26,27]. Live germinated conidia are referred to as LGC and inactivated germinated conidia as IGC. Inactivation was confirmed by the absence of growth of *T. rubrum* IGC on solid Sabouraud dextrose agar.

### 2.3. Co-Culture Conditions

For co-culture, LGC and IGC were transferred to 25-cm^2^ cell culture flasks containing macrophages previously differentiated with PMA (1 × 10^6^ cells/mL), as described previously [19]. For standardization of the co-culture challenged with bacterial lipopolysaccharide (LPS), the cultures were incubated with different LPS concentrations (1 µg/mL, 500 ng/mL, and 250 ng/mL) for different periods. The co-culture flasks containing fungal elements were incubated in an oven for 24 h at 37 °C in a 5% CO_2_ atmosphere, as illustrated in Figure 1.

### 2.4. Electron Microscopy

The interaction of *T. rubrum* LGC with human macrophages was analyzed by electron microscopy (JEOL JEM 100CXII electron microscope). Initially, we performed the induction of the cells for their transformation into macrophages with PMA, and then we performed the co-culture with live *T. rubrum* conidia according to [20] which used 6-well plates and was then incubated for 24 h. The electron microscopy test was carried out according to [28].

### 2.5. LDH Assay for Assessing the Viability of THP-1 Macrophages Co-Cultured with T. rubrum LGC

To evaluate viability of the macrophage cell line during co-culture with *T. rubrum* LGC, the release of lactate dehydrogenase (LDH) was analyzed after different incubation periods (5, 8, 12, 24, and 48 h) [29] using the TOX7 kit (Sigma-Aldrich) according to the manufacturer protocol, as described previously [20].

### 2.6. Quantification of Cytokines

Proinflammatory cytokines were quantified in the supernatant of co-cultures of THP-1 macrophages with *T. rubrum* LGC and IGC. A culture of THP-1 alone was used as negative control.

Quantification was performed in duplicate through two independent experiments. First, IL-12 (31–8000 pg/mL), IL-8 (1–150 pg/mL), and TNF-α (16–2000 pg/mL) were quantified using the ELISA Development Mini TMB EDK kit (PeproTech) according to manufacturer recommendations, as described previously [19]. After the transcriptome experiment, IL-32 (78.1–5000 pg/mL) was quantified using the R&D kit according to manufacturer recommendations.

### 2.7. RNA Isolation and Integrity Analysis

Total RNA was extracted using the miRNeasy^®^ Mini kit (Qiagen, Germany) according to the manufacturer’s instructions. After extraction, the absence of proteins and phenol in the RNA was confirmed in a MidSci Nanophotometer (Midwest Scientific, St. Louis, MO, USA) and RNA integrity was assessed by microfluidic electrophoresis in an Agilent 2100 Bioanalyzer (Agilent Technologies, Santa Clara, CA, USA). Only RNA with an RNA integrity number (RIN) >7.0 was used. These RNAs were quantified in a Quantus™ Fluorometer (Promega Corporation, Madison, WI, USA) to verify if they had the adequate concentration for library construction.

### 2.8. Library Construction and Sequencing

The cDNA libraries for RNA sequencing were constructed in triplicate for each condition (macrophages cultured with *T. rubrum* IGC and only macrophages as control). The libraries were obtained by paired-end sequencing in a HiSeq 2000 sequencer using the Illumina NextSeq kit, as described by [25]. The RNA-seq data are deposited in the Gene Expression Omnibus (GEO) database [29] under accession number GSE153327.

Only inactivated conidia were used during co-culture to avoid contamination of the macrophage culture with fungal mRNA.

### 2.9. Analysis of Sequencing Data

The raw RNA-seq data were filtered for quality control of the reads using FastQC (https://www.bioinformatics.babraham.ac.uk/projects/fastqc/, accessed on 3 May 2019) and trimmed with Trimmomatic [29] to remove adapters and other specific Illumina sequences from the reads. The paired-end trimmed reads of each sample were then aligned to the human hg38 genome using STAR aligner [30]. Read counts at the gene level were obtained using the quantMode GeneCounts option of STAR. Differential expression was analyzed with the DESeq2 Bioconductor package [31]. The Benjamini–Hochberg adjusted *p*-value threshold denoting statistical significance of changes in the expression levels of a given gene was set at 0.05, with a variation of ±1.0 being considered a significantly altered level of transcript abundance. Genes above these thresholds are referred to as differentially expressed genes (DEG). Gene ontology (GO) enrichment analysis of DEG was performed using the FunRich tool (http://www.funrich.org/, accessed on 3 May 2019).

### 2.10. QPCR Validation

A set of 16 genes were selected for qPCR validation. cDNA conversion and qPCR were performed as described previously [25] and gene expression levels were calculated using the 2^−ΔΔCT^ comparative method.

GAPDH [32] and β-actin [33] were used as normalizer genes for human macrophages, with the result being reported as mean ± standard deviation of three assays. Pearson’s correlation test was used to assess the correlation between qPCR and RNA-seq. The primers used for qPCR validation are given in Appendix A.

The genes for qPCR validation were selected in order to identify differences in gene expression between live and inactivated *T. rubrum* conidia. The same genes were used to compare the expression profile of genes modulated during co-culture of macrophages stimulated with *T. rubrum* and with bacterial LPS.

## 3. Results

### 3.1. Electron Microscopy of Co-Cultures

Electron microscopy analysis showed that macrophages cells engulf the *T. rubrum* conidia (Figure 2A). Additionally, we observed conidia particles encompassed by macrophages, demonstrating the formation of phagolysosome, starting cellular digestion. This process is essential for providing immune protection against pathogens (Figure 2B).

### 3.2. LDH Assay for Assessing the Viability of THP-1 Macrophages Co-Cultured with T. rubrum LGC

We performed the LDH assay at different times in order to evaluate the viability of macrophages after contact with LGC. The percentage of LDH release was 20% after 24 h; demonstrating that about 80% of the macrophages were viable and adequate for the subsequent assays. There was little interaction between the two organisms at the other time points (5, 8, and 12 h) (Figure 3)

### 3.3. RNA-Seq Analysis of Macrophages Co-Cultured with T. rubrum IGC

Sequencing generated a mean number of 90 million raw reads. Low-quality reads were removed. An average 90% of the high-quality reads were aligned to the hg38 (*Homo sapiens*) reference genome (UCSC Genome Bioinformatics site, Santa Cruz, CA, USA). Appendix A shows the total number of filtered and aligned reads.

### 3.4. Analysis of the Transcriptional Profile of Differentially Expressed Genes

In total, 83 genes Appendix A were differentially expressed during 24 h of co-culture. Table 1 shows the 10 most significantly modulated genes.

According to the distribution of the genes, a log2 fold change cut-off +/- 1 was established to define the most DEG.

### 3.5. Functional Categorization of Differentially Expressed Genes

The DEG were grouped by functional categories: biological processes, molecular functions, and cellular components. Among the functional category of biological processes, we chose to validate and discuss genes related to the immune response and other functions (Table 2).

Based on these categories, the genes selected for validation followed the criterion of highest fold change, with most genes being related to the immune response. Primers targeting the selected genes used for qPCR analysis are described in Appendix A.

### 3.6. Validation by qPCR

Pearson’s correlation coefficient was determined to evaluate the degree of correlation between the techniques used (RNA-seq and qPCR) for expression analysis of the 16 validated genes (Figure 4). There was a strong correlation (R = 0.98) between the RNA-seq and qPCR results.

### 3.7. Comparison of the Expression Profile of Validated Genes between Co-Cultures Using IGC and LGC

The validated genes were used to evaluate the existence of differences in the expression profile between co-cultures with IGC and LGC. Figure 5 shows a higher expression level of all macrophage genes evaluated (except for TRL8 and CD1D) when co-cultured with LGC compared to IGC. However, there was no difference in the profile of gene modulation (induction or repression) between the two conditions studied.

### 3.8. Analysis of the Release of Interleukins during Co-Culture

To evaluate the response of macrophages to IGC, we quantified the interleukins TNF-α, IL-8, and IL-12. We observed an increase in IL-12 release when macrophages were co-cultured with IGC compared to control (only THP-1) (Figure 6). There was no difference between groups in the other interleukins quantified.

The results of RNA-seq showed induction of the gene encoding IL-32. Thus, in order to assess the modulation of the protein encoded by this gene, we quantified the release of this interleukin during co-culture with IGC. As can be seen in Figure 7, there was a significant increase in the release of IL-32 in co-culture with *T. rubrum* IGC when compared to control.

### 3.9. Comparison of the Response of THP-1 Human Macrophages Co-Cultured with T. rubrum and Stimulated with Bacterial LPS

To assess whether genes modulated by macrophages were specific for a fungal response, we evaluated the same genes selected from the RNA-seq data in a co-culture of bacterial LPS-challenged human macrophages. First, the cells were co-cultured with different concentrations of LPS (1 µg/mL, 500 ng/mL, and 250 ng/mL) for 24 h. However, there was no modulation of any of the selected genes, including the TLR4 gene, which is a widely known LPS receptor.

We therefore constructed gene expression curves using different incubation times (3, 6, and 9 h) and concentrations of LPS (500 and 250 ng/mL). Modulation of all genes was observed at 9 h only. This period was chosen to evaluate the effect of LPS at a concentration of 500 ng/mL on THP-1 cells and to compare their response to co-culture with *T. rubrum* for 9 h (Figure 8).

## 4. Discussion

Currently, the number of patients with superficial infection caused by *T. rubrum* is increasing. Furthermore, the incidence of new cases of deep infections caused by this dermatophyte has encouraged the search for new drugs and new types of treatment by the scientific community [10,25].

The mechanisms underlying deep infections caused by *T. rubrum* have not yet been clarified. Within this context, molecular studies can be used to identify new target genes that could lead to the discovery of new drugs for treating and/or preventing the aggravation of superficial and deep infections. We highlight the importance of studies using different models of infection, which provide tools that can help elucidate the immune system responses of patients with deep infections caused by this dermatophyte.

To analyze the pathogen–host relationship, we co-cultured THP-1 human macrophages with germinated *T. rubrum* conidia. Electron microscopy revealed the internalization of conidia by THP-1 cells, demonstrating the interaction between the two organisms. These results agree with the literature showing that mouse macrophages were able to engulf *T. rubrum* conidia, which developed into hyphae inside the macrophages, bypassing the immune system [34].

After assessing the interaction between organisms, we evaluated the viability of macrophages after different periods of incubation (5–48 h) with *T. rubrum* LGC by quantifying the release of LDH. An incubation time of 24 h was chosen in the assays because about 80% of macrophages were viable after this period.

Ref. [25] showed a percent release of LDH of 18% during *Trichophyton rubrum* and HaCat Keratinocyte co-culture, demonstrating that 82% of human keratinocytes were viable after 24 h of interaction. In terms of co-culture condition [19], we evaluated the viability of THP-1 macrophages in response to inactivated *T. rubrum* and found that 70% of the cells were viable after 24 h of interaction. In addition, these authors demonstrated that the viability of macrophages decreased within 48 h of co-culture, in agreement with the results of the present study.

Based on a previous study from our research group [19], IL-6 was quantified in co-cultures of THP-1 cells with *T. rubrum* LGC at different incubation times in order to determine the best exposure time of the two cell types for the quantification of other interleukins and for use as a parameter of the immune response in the subsequent assays. According to [35], an increase in IL-6 release is an indicator of activation of the immune response induced by contact with *T. rubrum* and we, thus, showed activation of the immune system via IL-6. In addition, the LDH assay indicated an incubation time of 24 h as the most appropriate.

In the present study, we quantified the release of IL-12 during co-culture. This interleukin is induced by microorganism particles in monocytes and macrophages [36]. In addition, IL-12 plays an important role during systemic infection but is less important during skin infections [26]. The release of IL-12 was significantly greater during co-culture, showing that THP-1 cells were sensitized by inactivated conidia. This interleukin plays an important role in controlling fungal infection caused by *T. rubrum* since [36] showed that cells of IL-12-deficient mutant mice had a low phagocytic index and the fungal load on the infected skin was increased.

Our work demonstrated an association of IL-32 release with fungal infection caused by *T. rubrum*. Several studies have reported the action of IL-32. This interleukin controls the development of infection by inducing the expression of a cascade of proinflammatory cytokines [37,38]. The silencing of the gene encoding IL-32 in THP-1 macrophages infected with *Leishmania* led to an increase in infection [39]. Recently, IL-32 release was detected in oral lesions of patients infected with *Paracoccidioides* spp. [40].

Next-generation sequencing of the response of macrophages to co-culture with *T. rubrum* IGC revealed 83 modulated genes; of these, 65 were induced and 18 were repressed. The categorization of modulated genes showed their involvement in signal transduction, cell communication, and immune response pathways. Sixteen genes were selected for validation and Pearson’s correlation coefficient was 0.98, indicating a high correlation between RNA-seq and qPCR. We describe some modulated genes in the present study and, although not directly related to infections caused by *T. rubrum*, we highlight the importance of modulation of these genes during co-culture of THP-1 macrophages with *T. rubrum*, a tool that can be used to simulate the response to deep infection caused by this dermatophyte.

Based on the transcriptome results, we describe unpublished data that can be explored in an attempt to identify biomarker genes and possible therapeutic targets for the control of dermatophyte infection. We highlight the ANKRD1 gene, which was found to be induced in hepatitis C virus infection [41]. The silencing of this gene led to an increase in the viral load of herpes simplex virus [42]. The increased expression of the ANKRD1 gene was observed in skin lesions (important for wound healing and closure) [43]. The gene S1PR1 was induced in co-cultures of THP-1 cells with *T. rubrum* IGC for 9 h. Sphingosine-1-phosphate (S1P) and its receptor S1P receptor 1 (S1PR1) are potential therapeutic targets and biomarkers for sepsis. *S1PR1*-associated biomarkers could predict the survival of patients with sepsis using gene expression profiles of peripheral blood [44]. Moreover, the S1PR1 gene has been correlated with a reduction in acute lung injury induced by influenza H1N1 [45] and plays an important role in the physiopathology of *P. aeruginosa* [46].

The same validated genes of RNA-seq were used to determine the existence of differences in the expression profile between co-cultures with IGC and LGC. All macrophage genes evaluated (with the exception of TRL8 and CD1D) exhibited higher expression levels when LGC were used; however, we found no difference in the modulation profile (induction or repression) between the two conditions studied.

We compared the expression profile of genes selected by RNA-seq during co-culture macrophages with inactivated fungal elements of *T. rubrum* and macrophages challenged with LPS. Lipopolysaccharide is a major component of the wall of Gram-negative bacteria and is a classical activator of inflammation that is used in many inflammatory models. It is well established that after binding to the Toll-like receptor 4 (TLR4) on the membrane of immune cells, LPS rapidly triggers the activation of signaling pathways, inducing mitogen-activated protein kinase (MAPK) and nuclear factor-κ light-chain-enhancer of activated B cells (NF-κB) through MyD88-dependent and independent mechanisms and the subsequent expression of inflammatory cytokines by macrophages [47,48,49].

Analysis of CRLF2, MMP10, ANKD1, and SLC43A2 gene expression showed the same modulation profile in co-cultures of macrophages with LPS and with fungal elements of *T. rubrum*. On the other hand, the CSF1, CXCL1, CXCL2, CXCL3, CXCL8, IL-32, TLR7, and CD1D genes were induced by LPS but were repressed in co-culture with *T. rubrum*. The FCGBP and S1PR1 genes were repressed by LPS and induced in co-culture with *T. rubrum*. The TLR7 gene was up-regulated in the presence of LPS and was repressed in the presence of *T. rubrum*.

In recent years, the role of intracellular pattern recognition receptors (TLR3, TLR7, TLR8, and TLR9) has become increasingly important in the pathophysiology of some mycoses, such as paracoccidioidomycosis, cryptococcosis, aspergillosis, and candidiasis [50]. We observed repression of the TLR7 gene during co-culture of THP-1 cells with inactivated *T. rubrum* for 9 h, suggesting that the cellular response to *T. rubrum* may not be dependent on TLR7.

Most changes in gene expression are not yet clear but indicate a broad interaction between macrophages and *T. rubrum*. The identification of genes with specific activity against bacteria or fungi can be used to treat infectious and inflammatory diseases.

## 5. Conclusions

The co-culture of a macrophage cell line with fungal elements of *T. rubrum* proved to be an effective tool to understand the fungus–macrophage relationship. In particular, the increase in the level of the inflammatory interleukin IL-32 is an indicator of activation of the response elicited by the contact of macrophages with *T. rubrum*. The modulation of the expression of genes, such as IL-32, IL-8, and CSF-1, during co-culturing revealed new targets that can be used in therapies designed to increase the host immune response in deep infections caused by *T. rubrum*.

## Figures and Tables

**Figure 1 jof-09-00563-f001:**
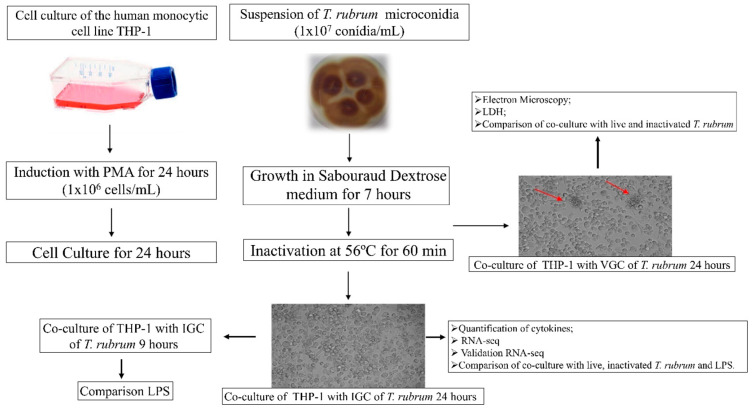
Co-culture of macrophages with live germinated (LGC) and heat-inactivated germinated (IGC) *T. rubrum* conidia. The arrows indicate the interaction between macrophages and fungal elements.

**Figure 2 jof-09-00563-f002:**
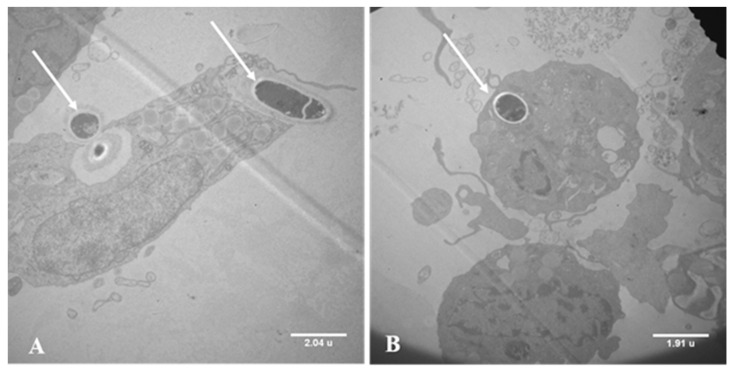
Co-culture of live germinated *Trichophyton rubrum* conidia (LGC) with the THP-1 human macrophage cell line for 24 h. (**A**) Macrophages cells engulf the *T. rubrum* conidia. (**B**) Formation of phagolysosome. Magnification: 5k×. The arrows indicate LGC inside macrophages.

**Figure 3 jof-09-00563-f003:**
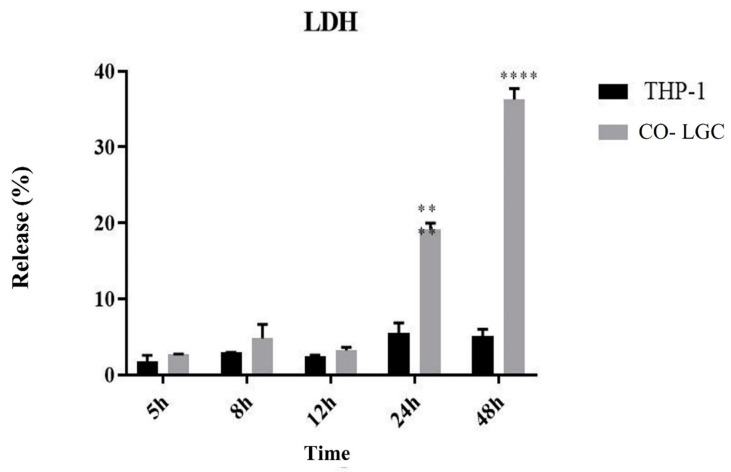
Release of LDH during co-culture of live germinated *T. rubrum* conidia (Co-LGC) with the THP-1 human macrophage cell line for 5, 8, 12, 24, and 48 h. Paired *t*-test. ** *p* < 0.01; **** *p* < 0.0001.

**Figure 4 jof-09-00563-f004:**
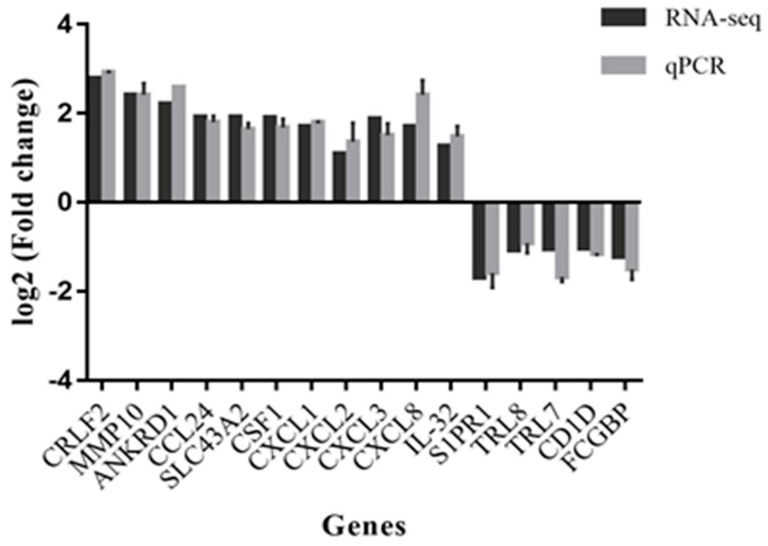
Comparison of the gene modulation profile found by RNA-seq and qPCR (R = 0.98, *p* < 0.001).

**Figure 5 jof-09-00563-f005:**
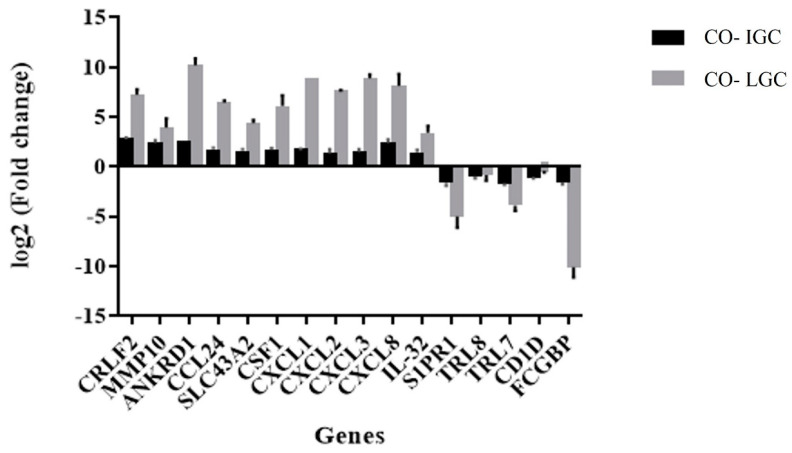
Comparison of co-cultures with live (Co-LGC) and inactivated (Co-IGC) germinated *T. rubrum* conidia.

**Figure 6 jof-09-00563-f006:**
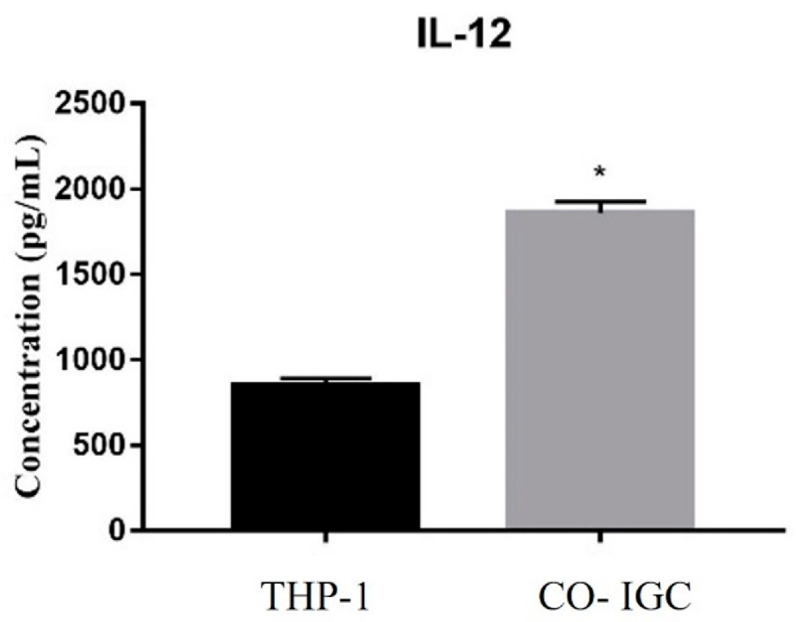
Quantification of IL-12 release (pg/mL) after 24 h. THP-1: only macrophages; CO- IGC: co-culture of THP-1 with inactivated germinated *T. rubrum* conidia. Paired *t*-test. * *p* < 0.001.

**Figure 7 jof-09-00563-f007:**
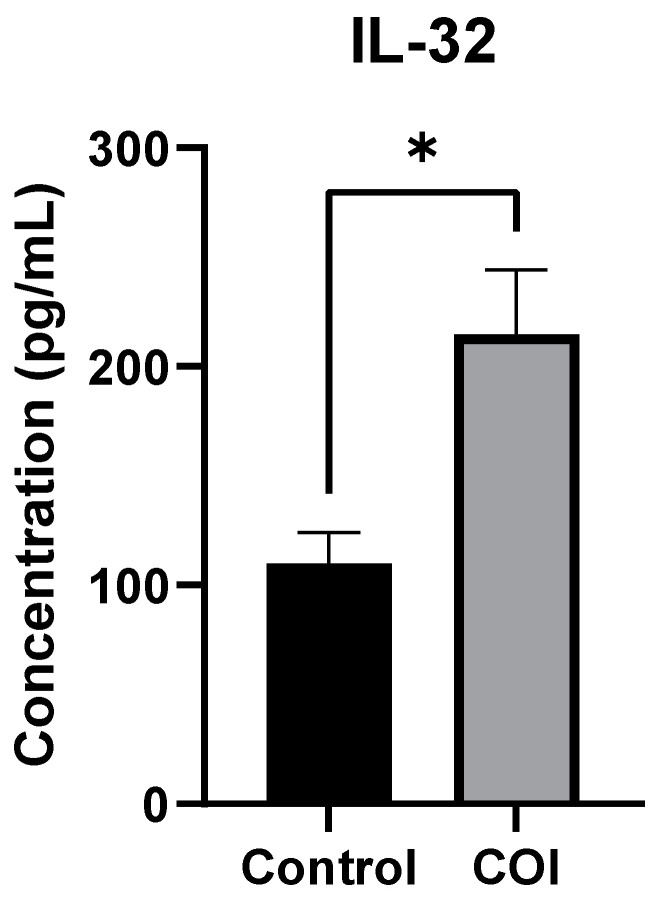
Quantification of IL-32 release after 24 h. THP-1: only macrophages; CO-I: co-culture of THP-1 with inactivated germinated *T. rubrum* conidia. Paired *t*-test. * *p* < 0.04.

**Figure 8 jof-09-00563-f008:**
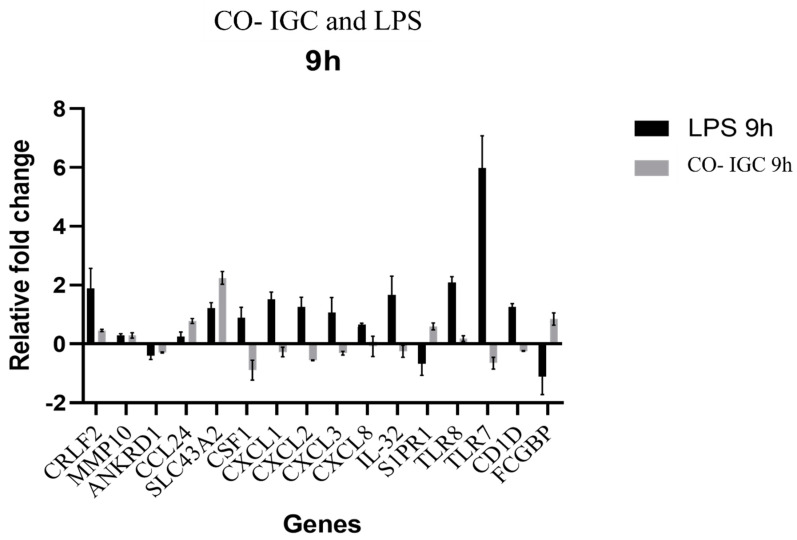
Comparison of the gene modulation profile of THP-1 macrophages co-culture challenged with LPS and co-cultured with inactivated germinated *T. rubrum* conidia (Co-IGC) for 9 h.

**Table 1 jof-09-00563-t001:** The top 10 most significantly up- or down-regulated genes.

ID	Gene Product Name	Log_2_ (Fold Change)
Up-regulated		
CRLF2	cytokine receptor-like factor 2	2.79
GREM1	gremlin 1, DAN family BMP antagonist	2.75
EBF1	EBF transcription factor 1	2.59
MMP10	matrix metallopeptidase 10	2.41
ANKRD1	ankyrin repeat domain 1	2.21
LIF	LIF interleukin 6 family cytokine	2.14
HIVEP2	human immunodeficiency virus type I enhancer binding protein 2	2.12
CCL24	C-C motif chemokine ligand 24	1.92
SLC43A2	solute carrier family 43 member 2	1.92
CSF1	colony stimulating factor 1	1.90
Down-regulated		
P2RY12	purinergic receptor P2Y12	−1.88
S1PR1	sphingosine-1-phosphate receptor 1	−1.69
NTS	neurotensin	−1.67
BBOX1	gamma-butyrobetaine hydroxylase 1	−1.44
LINC01537	long intergenic non-protein coding RNA 1537	−1.31
NLRP12	NLR family pyrin domain containing 12	−1.26
HPSE	heparanase	−1.24
FCGBP	Fc fragment of IgG binding protein	−1.22
PDE7B	phosphodiesterase 7B	−1.20
MMRN2	multimerin 2	−1.19

**Table 2 jof-09-00563-t002:** Validated genes involved in immune response and other functions of the biological processes category.

Biological Processes		
Immune response		
ID	Gene name	Log_2_ (fold change)
CRLF2	cytokine receptor-like factor 2	2.79
CCL24	chemokine (C-C motif) ligand 24	1.92
CSF1	colony stimulating factor 1 (macrophage)	1.9
CXCL3	chemokine (C-X-C motif) ligand 3	1.88
CXCL1	chemokine (C-X-C motif) ligand 1 (melanoma growth stimulating activity, alpha)	1.72
CXCL8	chemokine (C-X-C motif) ligand 8	1.71
IL32	interleukin 32	1.27
CXCL2	chemokine (C-X-C motif) ligand 2	1.10
CD1D	CD1d molecule	−1.03
FCGBP	Fc fragment of IgG binding protein	−1.22
Other functions		
MMP10 Protein metabolism	matrix metallopeptidase 10	2.41
ANKRD1 Regulation of nucleobase, nucleoside, nucleotide and nucleic acid metabolism	ankyrin repeat domain 1 (cardiac muscle)	2.21
SLC43A2 Transport	solute carrier family 43 member 2	1.92
S1PR1 Cell communication Signal transduction	sphingosine-1-phosphate receptor 1	−1.69
TLR8 Innate immune response	Toll-like receptor 8	−1.06
TLR7 Cell communication Signal transduction	Toll-like receptor 7	−1.04

## Data Availability

Not applicable.

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
