# Peer review of "Transcriptome Analysis of Co-Cultures of THP-1 Human Macrophages with Inactivated Germinated Trichophyton rubrum Conidia"

_jof, 2023, doi:10.3390/jof9050563_

Round 1

Reviewer 1 Report

The English needs to be modified as marked. 

Author Response

Reviewer Answers

Referee 1

 Author’s answer : We agree with suggestions and the manuscript was modified as described below

Patients with a weak immune system often develop deep dermatophytosis and, although less frequent, this infection can be fatal. Once the infection is established, the pathogen is able to reach deep layers, such as the dermis and hypodermis, and spread through the lymph nodes and bloodstream [10].

These infections are also reported in patients with a compromised immune system such as HIV carriers and patients with diabetes mellitus [9, 11, 12], demonstrating the importance of the immune system in controlling these pathologies. Studies in the literature have shown immunocompromised patients with recurrent dermal abscesses caused by fungi [13]. [14].

In view of the problems associated with deep infections and the need of discovering new types of treatment in order to control  these infections, it is necessary to understand the immunological mechanisms involved in this infection [15]

Therefore, the aim of the present study to perform molecular characterization of deep infection caused by T. rubrum using as a tool co-culture with a human macrophage cell line (THP-1).

2.2 Trichophyton rubrum strain, media, growth conditions and inactivation

Trichophyton rubrum CBS 118892 (CBS-KNAW Fungal Biodiversity Center), sequenced by the Broad Institute (Cambridge, USA), was cultured on Sabouraud dextrose agar (Oxoid, Hampshire, England) for 15 days at 28ºC. Conidia were prepared as described previously by [23]. A solution of T. rubrum (1 × 107 conidia/mL) was cultured in 5 mL liquid Sabouraud medium for 7 h under gentle shaking, as described [24, 25]. After this period, the fungal material was centrifuged for 10 min at 4,000 rpm. Part of these live conidia were stored and the other part was washed with sterile saline and incubated for 60 min at 56ºC for inactivation [26, 19]. Live germinated conidia are referred to as LGC and inactivated germinated conidia as IGC. Inactivation was confirmed by the absence of growth of T. rubrum IGC on solid Sabouraud dextrose agar (data not shown).

Reviewer 2 Report

The authors describe the interaction of human macrophages with Trichophyton rubrum conidia. Especially, the transcriptome of macrophages were analyzed, using inactivated T. rubrum microconidia. Identification of up and down regulated human genes were presented and results confirmed with qPCR methods.

Minor points.

Material and Methods:

2.4. Electron microscopy

What is about pretreatment of cells for electron microscopy? The authors should include citation of the used method.

Line 187 Table S1: supplemental material was not up loaded.

The authors should read the instructions for authors and include information about supplementary data in the main text.

Results:

Lines 194: “T. rubrum conidia (LGC) were able to penetrate...” Why the authors draw the conclusion that the conidia penetrates the macrophage? The normal process is that the macrophage engulf the fungal conidia. The authors should indicate the structures seen in Figure 2 in more detail. What happen in phagolysosome formation of macrophages?

Line 216 “TableS2”

Line 220 “Figure S1” as before, material was not up loaded.

Author Response

Referee 2

Open Review

Minor points.

Material and Methods:

2.4. Electron microscopy

What is about pretreatment of cells for electron microscopy? The authors should include citation of the used method.

Author’s answer  the information about pretreatment of cells for for electron microscopy  and citation were included

Line 187 Table S1: supplemental material was not up loaded.

The authors should read the instructions for authors and include information about supplementary data in the main text.

Author’s answer:  The tables S1, S2 and S3 were added in supplemental material

Results:

Lines 194: “T. rubrum conidia (LGC) were able to penetrate...” Why the authors draw the conclusion that the conidia penetrates the macrophage? The normal process is that the macrophage engulf the fungal conidia. The authors should indicate the structures seen in Figure 2 in more detail. What happen in phagolysosome formation of macrophages?

Author’s answer: The manuscript was correct.

 Electron microscopy analysis showed that macrophages cells engulf the T. rubrum conidia (Figure 2A). Addition, we observed conidia particles emcompassed by macrophages, demonstrating the formation of phagolysosome, starting cellular digestion. This process is essential for providing immune protection against pathogens. (Figure 2B).  .

Line 216 “TableS2”

Line 220 “Figure S1” as before, material was not up loaded.

Author’s answer:  The tables S1, S2 and S3 were added in supplemental material

Reviewer 3 Report

I am very glad to review this manuscript. I have some comments for this manuscript.

1.     the aim of the present study was to perform cellular and molecular characterization of deep infection caused by T. rubrum using as a tool co-culture with a human macrophage cell line (THP-1).  I think this manuscript mainly focus the Transcriptome analysis of co-cultures of THP-1 human macrophages. So, I do not agree the cellular mechanism.

2.     Figure 1 should be retypesetting. What dose VGC in figure 1 mean?

3.     Line 302: The sentence is not complete.

4.     Figure 7 whether the difference is of significance should be labelled in the figure.

5.     Reference style should be consistent. Reference title should not be in upper and lower case.

6.     The most important finding of the manuscript is the role of IL-32. Endogenous IL-32 mediates the production of TNF-α, IL-1β, IL-6, and IL-8 upon stimulation of THP-1 cells with LPS or PMA. So, I suggest the author should supply the results of THP-1 stimulated by LPS. It is better to explore and discuss the correlation of IL-32 signal transduction.

Author Response

Referee 3

I am very glad to review this manuscript. I have some comments for this manuscript.

Author’s answer:  thanks for suggestions!

  1. the aim of the present study was to perform cellular and molecular characterization of deep infection caused by T. rubrum using as a tool co-culture with a human macrophage cell line (THP-1).  I think this manuscript mainly focus the Transcriptome analysis of co-cultures of THP-1 human macrophages. So, I do not agree the cellular mechanism.

Author’s answer: Thanks for suggestion, We agree with your suggestion.  The manuscript was modified. The aim of this study was to analyze the transcriptome of a human monocyte/macrophage cell line (THP-1) co-cultured with inactivated germinated T. rubrum conidia (IGC) in order to characterize deep infection (lines 22-24)

  1. Figure 1 should be retypesetting. What dose VGC in figure 1 mean?

Author’s answer: the figure 1 was modified, sorry,  there was a mistake in tradution portugues-inglish:   VGC means LGC (live germinated conidia)

  1. Line 302: The sentence is not complete.

Author’s answer: The authors declare that the research was conducted in the absence of any commercial or financial relationship that could be construed as a potential conflict of interest.

This information was added in the manuscript.

  1. Figure 7 whether the difference is of significance should be labelled in the figure.

Author’s answer: The information was added in figure 7

Figure 7. Quantification of IL-32 release after 24 h. THP-1: only macrophages; CO-I: co-culture of THP-1 with inactivated germinated T. rubrum conidia. Paired t-test. *p<0.04.

  1. Reference style should be consistent. Reference title should not be in upper and lower case.

Author’s answer: The reference style was revised.

  1. The most important finding of the manuscript is the role of IL-32. Endogenous IL-32 mediates the production of TNF-α, IL-1β, IL-6, and IL-8 upon stimulation of THP-1 cells with LPS or PMA. So, I suggest the author should supply the results of THP-1 stimulated by LPS. It is better to explore and discuss the correlation of IL-32 signal transduction.

Author’s answer: The results of RNA-seq showed induction of the gene encoding IL-32. Thus, in order to assess the modulation of the protein encoded by this gene, we quantified the release of this interleukin during co-culture with IGC. As can be seen in Figure 7, there was a significant increase in the release of IL-32 in co-culture with T. rubrum IGC when compared to control.

Round 2

Reviewer 3 Report

My previous suggestion was not responded.

1.     Reference style should be consistent. Reference title should not be in upper and lower case. Name of fungi should be italicized in the title of references.

2. Line 304: The sentence is not complete. The sentence should be rewritten.

Author Response

  1. Reference style should be consistent. Reference title should not be in upper and lower case.

Author’s answer: The reference style was revised.

2. Line 304: The sentence is not complete. The sentence should be rewritten.

Author’s answer

Sorry, but  in my version the line 304 was complete (transcribed below) . Please send me a sentence that should be rewritten. 

[25] showed a percent release of LDH of 18% during Trichophyton rubrum and HaCat Keratinocyte co-culture, demonstrating that 82% of human keratinocytes were viable after 24 h of interaction. In co-culture condition [19] evaluated the viability of THP-1 macrophages in response to inactivated T. rubrum and found that 70% of the cells were viable after 24 h of interaction. In addition, these authors demonstrated that the viability of macrophages decreased within 48 h of co-culture, in agreement with the results of the present study.

Author’s answer

Please, the english of manuscript  was revised . The other two reviewers had no problem  with English understand. 
